# Extracellular Vesicle microRNAs as Possible Liquid Biopsy Markers in HNSCC—A Longitudinal, Monocentric Study

**DOI:** 10.3390/cancers16223793

**Published:** 2024-11-11

**Authors:** Carla Apeltrath, Frank Simon, Armands Riders, Claudia Rudack, Maximilian Oberste

**Affiliations:** Department of Otorhinolaryngology, Head and Neck Surgery, University Hospital Münster, 48149 Münster, Germany; capeltra@uni-muenster.de (C.A.); frank.simon@ukmuenster.de (F.S.); armands.riders@ukmuenster.de (A.R.); rudackc@ukmuenster.de (C.R.)

**Keywords:** liquid biopsy, biomarkers, microRNA, extracellular vesicles, head and neck squamous cell carcinoma (HNSCC), HPV, oropharyngeal carcinoma, oncology

## Abstract

Head and neck squamous cell carcinoma (HNSCC) consists of a group of heterogeneous tumors that are diagnosed comparatively late and often have a poor prognosis. Current head and neck research is motivated to find biomarkers for early detection, diagnosis, and therapy monitoring. Extravesicular (EV)-microRNA obtained by liquid biopsy of blood offers a promising approach here since the respective target mRNAs and their function in tumor neogenesis are partly already known. In this study, blood sera from 50 HNSCC patients and 16 controls treated between 2020–2023 at the Department of Otorhinolaryngology, Head and Neck Surgery of the University Hospital of Münster, were obtained by liquid biopsy. A specific EV-miRNA panel consisting of EV-miR-21, -1246, -200c, -let-7a, -11a, and -26a was determined. The expression of the analyzed EV-microRNA of the patients was then examined for correlation with clinical parameters regarding the diagnostic and prognostic value and suitability for therapy monitoring.

## 1. Introduction

Head and neck squamous cell carcinomas (HNSCCs) are one of the seven most common tumors, with incidence and mortality varying in different regions of the world [1]. The combination of high regular alcohol and tobacco consumption increases the risk of developing HNSCC 14-fold compared to non-smokers and non-alcoholics [2]. On the other hand, there is another group of HNSCC associated with high-risk human papilloma virus (HPV), which has now become the second main risk factor for HNSCC in general and can be detected in more than half of cases of oropharyngeal squamous cell carcinoma (OPSCC) [3]. HPV-positive OPSCC is a distinct disease entity compared to HPV-negative OPSCC, exhibiting a more favorable response to treatment and better overall survival (OS) [4].

Nevertheless, about one-third of patients diagnosed with and treated for locally advanced HNSCC develop a recurrence of the tumor in the follow-up period [5]. Taken together, markers for HNSCC could help in diagnosis and tumor after-care and potentially reduce mortality of the malignancy.

Up to today, reliable biomarkers for the early detection of HNSCC are lacking. It is crucial to gain a more in-depth understanding of the molecular mechanisms that underlie and characterize HNSCC to improve disease outcomes through advanced molecular diagnostics and more effective treatments. Recent findings have identified microRNAs within extracellular vesicles (EV-miRNAs) as significant regulatory molecules in HNSCC. These EV-miRNAs present promising potential as biomarkers for the disease and as new therapeutic targets [6].

MicroRNA are RNA molecules with a length of ~22 nucleotides, which have no genetic coding function but interfere with gene regulation at the mRNA level. They can prevent translation through mRNA binding or lead to its degradation [7].

Extracellular vesicles (EVs) are particles enclosed by a phospholipid bilayer, released by a variety of cell types, and present in blood, saliva, and other bodily fluids. EVs are categorized into exosomes (30–150 nm, originating from endosomes), microvesicles (100–1000 nm, originating from the plasma membrane), and apoptotic bodies (50–5000 nm, also derived from the plasma membrane), distinguished by their origin, size, and surface molecules [8]. They carry proteins, mRNA, non-coding RNA (such as microRNAs and long-noncoding-RNAs), DNA, and lipids, physiologically facilitating cell-to-cell communication. However, EVs are implicated in cancer progression, too. In patients with HNSCC, plasma is enriched in EVs, which can contribute to tumor progression in different ways, e.g., by promoting angiogenesis, metastasis, TME reprogramming, immune tolerance, and therapy resistance [9,10,11]. The terms “EV” and “exosome” are sometimes used synonymously. MISEV guidelines recommended cautious use of the term “exosome” and reaffirmed this in the update published in 2024 [12,13]. On this basis, the term “extracellular vesicle (EV) micro-RNA” is used in the following sections for the research presented by our working group. Results from other research groups that use the term exosomal miRNA, cited in this paper, remain unchanged.

MiRNAs in EVs can have an oncogenic effect on the recipient cell, delivered by tumor cells, but also by cells of the immune system and promote tumorigenesis and progression; at the same time, it has also been shown that immune cells can package and deliver tumor-suppressive microRNAs in EVs [9].

Different tumor-suppressive and oncogenic miRNAs have shown pathological up- or downregulation in HNSCC. We chose a panel of six EV-miRNAs (miR-21, -1246, -200c, -let-7a, -181a, -26a) that are already known to show comparatively sensitive dysregulated EV expression in patients with HNSCC. In several studies, EV-miR-21 overexpression correlates with tumor stage, lymph node metastasis, and poorer prognosis in HNSCC. In addition, EV-miR21 also appears to have potential as a monitoring marker, as expression decreases after surgery in patients with a good prognosis, while patients with a poor prognosis continue to have high levels [10].

MiR-1246 is upregulated in tumor tissue as well as in the serum (exosomes) of patients with HNSCC of laryngeal origin [14]. Huang et al. were also able to demonstrate a correlation between high expression of miR-1246 in extracellular vesicles in patients with LSCC and a poorer prognosis. MiR-1246 is best studied in laryngeal HNSCC.

MiR-200c has diagnostic and prognostic value, it is downregulated in HNSCC tumor tissue compared to healthy controls [15], and patients with recurrence of laryngeal carcinoma have lower miR-200c levels in tumor tissue than those without recurrence [16]. In HPV-negative OPSCC specifically, however, upregulated miR-200c has already been detected in tumor tissue in vitro and in vivo [17].

MiR-let-7a belongs to the let-7 family, which are tumor-suppressive micro-RNAs and can be detected downregulated in the serum of patients with HNSCC [10,18]. Studies on miR-let-7a levels in serum are still rather limited, but let-7a also appears to be expressed lower in serum in laryngeal carcinomas; the miRNA causes an inhibition of LSCC proliferation and promotes apoptosis in in vitro experiments [19].

MiR-181a is considered tumor-suppressive and is downregulated in HNSCC [10]. It has also been shown that in vitro infection with HPV16 in (HPV-negative) OSCC cells increases tumor stem cell potential via downregulation of miR-181a [20]. However, EV-miRNA does not appear to have an exclusively tumor-suppressive effect in HNSCC; rather, it is increasingly characterized as “pleiomorphic”. In the plasma of OSCC patients, its serum expression correlated with TNM status and overexpression in the tissue of ESCC correlated with advanced TNM status [10].

MiRNA-26a has tumor-suppressive properties and is expressed at reduced levels in HNSCC tissue [10]. In HNSCC of oral origin, low expression correlated with poorer prognosis through negative correlation with tumor size and lymph node metastasis, while overexpression of miRNA in vitro promoted apoptosis and inhibited tumor cell progression [21]. Most of the findings relate to expression behavior in tissues, and their expression behavior in EVs seems to be poorly studied.

Current research on EV-miRNA and liquid biopsy and their differences in the different tumor locations of HNSCC is still sparse, and many studies consider a tumor entity or undifferentiated HNSCC in general. The aim of this study is to understand the dynamic of EV-miRNA expression, find miRNAs that correlate with parameters in diagnosis, and further qualify for therapy monitoring and even prognosis. Additionally, we want to explore the differences in EV-miRNA expression in HNSCC of different subsites with a focus on OPSCC and its HPV association.

## 2. Materials and Methods

### 2.1. Patient Acquisition

Patients for this study were recruited at the Department of Otorhinolaryngology, Head and Neck Surgery, University Hospital Münster, Germany. This study was approved by the local ethics committee of the Westphalia Lippe Medical Association and the Münster University of Applied Sciences (AZ 2021-210-f-S). When a malignant disease was suspected, the patient’s consent to participate in this study was obtained during the preparation for surgery, and the blood was then taken before surgery. If HNSCC was detected, depending on the staging and feasibility of surgery, primary radio-chemotherapy or primary surgery with adjuvant radio-chemotherapy or primary radiotherapy was planned and carried out based on the decision of an interdisciplinary tumor board. Follow-up care for HNSCC was carried out according to the standard at intervals of three months in the first year after diagnosis. For the follow-up examination (FU), blood was taken from the patients earliest at 6 months FU and at least twelve months after the initial diagnosis (<12 M FU). The control group consisted of 16 patients, eight male and eight female, that underwent surgery for either benign or suspected malign lesions that afterward were classified as benign lesions (e.g., fibromas, cysts) in the pathological examination.

A summary of the following steps in the lab can be seen in Figure 1.

### 2.2. Serum Preparation

Isolation of EVs and the characterization of the miRNA cargo were based on the minimal experimental guidelines of the International Society for Extracellular Vesicles (MISEV 2018) [12] at the time of the start of the investigations. Whole blood was collected in Sarstedt S-Monovettes, and serum was separated by centrifugation (1900× *g*, 10 min, 4 °C). Separated serum was centrifuged (3000× *g*, 15 min, 4 °C) to remove additional cellular nucleic acids attached to cell debris.

### 2.3. EV Isolation

A total of 500 µL of cleared serum was used for precipitation of exosomes and other EVs with polyethylene glycol using the miRCURY Exosome Serum/Plasma Kit (Qiagen, Hilden, Germany) according to manufacturer’s instructions. Demonstration of suitability of the kit regarding size, surface markers and quantity of recovered EVs is given by Helwa et al. [22]. Particle size recovered by the kit ranges between 40–150 nm.

### 2.4. Micro-RNA Isolation from EVs

Total RNA including miRNA was isolated from EVs using the miRNeasy Micro Kit (Qiagen, Hilden, Germany). In brief, isolated EVs including exosomes were lysed by the QIAzol method, and total RNA including miRNA was isolated using spin column-based centrifugation according to the manufacturer’s instructions. We added synthetic RNA Spike-ins (UniSp2, UniSp4, UniSp5) to the QIAzol lysis buffer to control the quality of RNA isolation, cDNA synthesis, and PCR amplification. A table listing the Ct-values of the investigated miRNAs and Uni-Spike-In controls can be found in the Appendix A. Isolated RNA was eluted from RNeasy MinElute Spin Columns with 14 µL RNAse-free water. Isolated RNA was of high purity (OD260/OD280 = 1.8–2.0) and moderate to high quality (RIN ≥ 6).

### 2.5. Reverse Transcription (cDNA Synthesis)

Isolated total RNA including miRNA was reversely transcribed using the miRCURY LNA RT Kit (Qiagen, Hilden, Germany) according to the miRCURY LNA miRNA SYBR Green PCR Biofluid Samples Handbook instructions. Because of limited RNA amount in serum, calculation of the volume of template RNA used for cDNA synthesis for a total reaction volume of 10 µL was as follows:

Template RNA [µL] = (elution volume [µL]/original sample volume used [µL] × 8 [µL]). In brief, the following components were added to 10 µL of reaction mixture: 2 µL 5 × miRCURY Reaction buffer, 1 µL miRCURY RT Enzyme Mix, 0.5 µL UniSP6 RNA Spike-in, 6 µL 1:10 diluted total RNA (~1.8 ng), and 0.5 µL RNAse-free water. cDNA synthesis was carried out at 42 °C for 60 min, following 95 °C for 5 min in a thermocycler. Synthetic UniSp6 RNA Spike-in was used in the reverse transcription reaction to check for cDNA synthesis efficiency.

### 2.6. Quantitative Real-Time PCR Amplification

Real-time PCR amplification of cDNA was carried out using miRCURY LNA SYBR Green PCR Kit (Qiagen, Hilden, Germany) according to the manufacturer’s instructions. The following miRCURY LNA miRNA PCR Assays (Qiagen, Hilden, Germany) were used for amplification of 8 miRNAs and 4 RNA Spike-in controls: hsa-miR-181a-5p (YP00206081), hsa-miR-26a-5p (YP00206023), hsa-let-7a-5p (YP00205727), hsa-miR-16-5p (YP00205702), hsa-miR-1246 (YP00205630), hsa-miR-200c-3p (YP00204482), hsa-miR-196a-5p (YP00204386), hsa-miR-21-5p (YP00204230), UniSp5 (YP00203955), UniSp4 (YP00203953), UniSp2 (YP00203950), UniSp6 (YP00203954).

The reaction mixture (10 µL) consisted of 5 µL 2 × miRCURY SYBR Green Master Mix, 0.5 µL 20 × ROX reference dye, 1 µL miRCURY LNA miRNA PCR assay, 3.0 µL 1:30 diluted cDNA Template, and 0.5 µL RNAse-free water. Real-time PCR was carried out on ABI 7900 HT thermal cycler at the following conditions: 95 °C for 2 min, followed by 40 cycles at 95 °C for 10 s and 56 °C for 1 min. Following PCR, a melting curve analysis was carried out to check for the presence of nonspecific amplification products. Each sample was analyzed in triplicate. As a suitable reference gene (endogenous control) for qPCR analysis of serum, miRNA-16 was chosen. MiRNA-16 is often used as endogenous reference in similar studies on miRNA in HNSCC, e.g., by Piao et al. in OSCC [23].

### 2.7. Data Analysis and Statistics

EV-miRNA levels were quantified by measuring the value of the cycle threshold (∆CT) with miR16 as an endogenous control (∆CT = mean CT_miRNA_ − mean CT_mir16_). *Fold change (FC) expression* of each miRNA was calculated using the 2^−∆∆CT^-method and presented as log_2_FC. IBM SPSS Statistics software, Version 28.0.0.0 (Armonk, NY, USA) was used for statistical analysis of the micro-RNA expression examined, and calculations were made using general linear models with two measurement times. *p* values < 0.05 were considered significant. To investigate the relationship between UICC stage, HPV status, tobacco use, and alcohol consumption, planned therapy and tumor location with the EV-microRNA load of the microRNAs miR-21, miR-1246, miR-200c, miR-let7a, miR-181a, and miR-26a at initial diagnosis of the tumor disease, general linear models with two measurement time points are used in this study. A prognostic model was created for each microRNA. As an example, a fictional patient based on the EV-miR-21 prognostic model with the following clinical parameters: laryngeal carcinoma, T1, N0, UICC I, HPV-negative, treated with surgery, tobacco consumption, and occasional alcohol consumption or active alcohol abuse would have the following Log_2_ expression ratio compared to the normal collective: 0.684 (see Appendix B, Table A1). Back-transformed (2^0.684^), this would be a 1.61-fold higher expression of EV-microRNA-21 in this fictional patient than the average expression of EV-microRNA-21 in the healthy cohort. Using paired *t*-tests, the expression ratios of the six miRNAs in relation to the clinical parameters examined at the time of surgery and at the follow-up (avg. 12 months) were analyzed for significant changes over this period.

## 3. Results

### 3.1. Description of the Patient Cohort

The total sample size of the tumor cohort comprises *n* = 50 patients. Of these, 24% (*n* = 12) are female patients, and 76% (*n* = 38) are male patients. The age range varies between 36 and 84 years; the mean age is M = 63.94 (SD = 10.23) years. In Table 1, clinical parameters of the staging of the tumor patients are listed. In Table 2, further aspects of the oncological anamnesis, HPV association, and recurrence situation are displayed.

### 3.2. Results of Interference Statistical Analysis of Clinical Parameters and EV-microRNA Expression

At the time of initial diagnosis, EV-miR-21 expression was 2.89 times higher (Log_2_ regression coefficient: 1.533, *p* = 0.005) in the group of patients with N0 stage (*n* = 25) compared to the group of patients with positive lymph node status (*n* = 22) (Figure 2a). In patients with HPV/p16 positive tumors (*n* = 14), the EV expression of EV-miR-21 was 2.88 times higher (Log_2_ regression coefficient: 1.525, *p* = 0.003) than in patients with HPV/p16-negative tumors (*n* = 33) (Figure 2b and Table A1). The expression ratios of the EV-miRNAs also were examined for a significant change over the 12-month period using paired *t*-tests. The EV expression of micro-RNA-21 showed a significant increase in patients with positive lymph node status (*p*_*t*-test_ = 0.032) and with HPV/p16 negative tumors (*p*_*t*-test_ < 0.001) over the 12-month follow-up period compared to the first liquid biopsy at initial diagnosis (Figure 2a,b).

For EV-microRNA-1246, the prognostic model also revealed significant differences between the lymph node involvement in the HNSCC tumor group. Patients without lymph node metastases (*n* = 25) showed 3.46 times higher (Log_2_ regression coefficient: 1.791, *p* = 0.035) EV-miR-1246 expression than patients with positive lymph node status (Figure 3a and Table A2). The group of patients with positive lymph node status (*p*_*t*-test_ = 0.029) also recorded a significant increase in miR-1246 EV expression over the observed period (Figure 3a). In the sub-analysis with the OPSCC patients, similar results could be observed. A negative lymph node status (*n* = 11) correlated significantly with a 7.8-fold higher (Log_2_ regression coefficient: 2.963, *p* = 0.002) expression of EV-miR-1246 compared to a positive lymph node status (*n* = 11) (Figure 3b and Table A3). The group with lymph node metastases (*p*_*t*-test_ = 0.049) showed a significant increase in EV-miR-1246 over the 12 months (Figure 3b). There was no significant result found regarding a correlation between EV-miR-21 and OPSCC-patients.

Looking at the HNSCC cohort, the HPV/p16 positive patients (*n* = 13) showed a 2.5-fold higher expression (Log_2_ regression coefficient: 1.321, *p* = 0.028) of EV-miR-let-7a (Figure 4a) and, for EV-miR-181a, the HPV/p16 positive (*n* = 14) showed a 2.91-fold higher (Log_2_ regression coefficient: 1.544, *p* = 0.005) expression than the HPV/p16 negative patients (*n* = 33) (Figure 4b, Table A4 and Table A5).

Taken together, EV-miR-21, -181a, and -let-7a are significantly higher expressed in HPV/p16+ HNSCC (Figure 5) at the time of initial diagnosis.

Furthermore, the type of therapy showed a significant correlation with the expression of EV-miR-181a at the time of the follow-up examination in the HNSCC cohort. In comparison to the patients treated with definitive combined radio-chemotherapy (*n* = 4), the group that received surgery and adjuvant radiotherapy (*n* = 10) showed 2.61 times higher (Log_2_ regression coefficient: 1.383, *p* = 0.02) EV expression levels of miR-181a at the time of follow-up (Figure 6a). Similarly, the patients that received only surgery (*n* = 20) showed 3.2 times higher (Log_2_ regression coefficient: 1.679, *p* = 0.008) EV expression levels of miR-181a in comparison to the group of patients that were treated with surgery and adjuvant radiotherapy (*n* = 10) at the time of follow-up (Figure 6a). Again, in the follow-up examination, patients who had been treated with surgery and adjuvant radiotherapy (*n* = 10) showed a 2.17 times higher expression (Log_2_ regression coefficient: 1.117, *p* = 0.044) and patients who had only been treated with surgery (*n* = 20) showed a 2.64 times higher (Log_2_ regression coefficient: 1.402, *p* = 0.017) expression of EV-miR-26a than patients who had received definitive radio-chemotherapy (*n* = 4) (Figure 6b and Table A6). Similar results were found for the OPSCC cohort. The expression levels of EV-microRNA-181a in the follow-up showed a significant correlation with the treatment modality: Patients who had been treated with surgery and adjuvant radio-chemotherapy (*n* = 7) had 2.92 times higher (Log_2_ regression coefficient: 1.546, *p* = 0.046) and patients treated with surgery and adjuvant radiotherapy (*n* = 9) had 3.04 times higher (Log_2_ regression coefficient: 1.604, *p* = 0.026) expression levels of EV-miR-181a compared to patients treated with definitive radio-chemotherapy (*n* = 2) (Figure 6c and Table A7). For micro-RNA-26a, similar significant correlations were found in the OPSCC patients with the form of therapy used. Compared to patients treated with definitive radio-chemotherapy (*n* = 2), the group treated with surgery and adjuvant radio-chemotherapy (*n* = 7) expressed EV-miR-26a 3.34 times higher (Log_2_ regression coefficient: 1.741, *p* = 0.035), and the group that had received surgery and adjuvant radiotherapy (*n* = 9) expressed EV-miR-26a 3.82 times higher (Log_2_ regression coefficient: 1.934, *p* = 0.013) (Figure 6d and Table A8).

No significant relation between disease-free survival nor recurrence of the HNSCC and the analyzed EV-microRNA could be found in our study.

## 4. Discussion

This study combines several current research approaches in oncological head and neck research: the method of liquid biopsy for non-invasive sampling and the examination of EV-miRNA. However, we do investigate one tumor localization alone, but for HNSCC in general and at the same time, we carry out a sub-analysis for OPSCC. The findings of this work shall complement and enrich the knowledge about EVs in HNSCC, differences in EV-miRNA expression depending on the localization in the head and neck region, and particularly the knowledge about the change in EV-miRNA expression during a follow-up examination.

We were able to show that when looking at EV-miRNA at baseline, significant differences were found in three miRNAs (EV-miR-21, -181a, -let-7a) with regard to HPV/p16 status.

The upregulation of both oncogenic and tumor-suppressive miRNA in HPV+ HNSCC is not necessarily a contradiction; HPV+ tumors have a much better prognosis [24], but they are regularly clinically apparent due to early lymphogenous metastasis [3]. Comparing HPV16+ cell lines, Iuliano et al. showed that miR-21 is downregulated in the cells but upregulated in the EVs they deliver [25]. Nevertheless, our findings underline that certain EV-miRNAs could serve as markers for HPV associated HNSCC.

Further, a significant increase in EV-miR-21 expression over 12 months was recorded in the group of HPV/p16 negative patients. MiR-21 seems to decrease after successful surgery for HNSCC, while the persistence of elevated levels indicates a rather unfavorable prognosis [26]. EV-microRNA-21 showed no significant correlation with the occurrence of recurrence in the follow-up period of about 12 months, but the significant increase in EV-miR-21 in the HPV/p16-negative group yet could correlate with the commonly poorer prognosis of these tumors and reflect the poorer OS after 40 months, the higher risk of progression, and the higher mortality compared to HPV+ HNSCC [27,28].

So far, only a reduced tissue expression of EV-miR-181a in vitro and in the examination of resected HPV-positive OPSCC have been described [10], but since EV expression can deviate from this, further studies should investigate EV-miR-181a and HPV status. MiR-let-7a and its expression behavior depending on HPV status is currently mostly investigated in studies with HPV-associated cervical carcinomas, while data regarding EV-miR-let-7a in HPV/p16 positive HNSCC are currently still lacking. Nevertheless, aligning with our results, free plasma concentrations of miR-let-7a are higher in patients with HPV+ cervical carcinomas than in HPV- patients [29].

EV-miR-21 and EV-miR-1246 were higher expressed in patients with N0-stage than in patients with positive lymph node status at initial diagnosis. However, for both EV-miR-21 and -1246, an increase in expression could be seen in the group of patients with lymph node metastasis in the follow-up examination. Despite the oncogenic nature of miR-21 mentioned in the literature and the exosomal miR-21 levels correlating with increased N-stage in studies with hypoxic OSCC [30], the results observed by us are in line with several studies that could not show a significant correlation between the expression of miR-21 in tumor tissue and the N-stage in OPSCC [31,32], as we could not find significant correlation of EV-miR-21 or N-stage in OPSCC. In the sub-analysis with OPSCC, again, the patients with N0-stage had higher EV expression levels of miR-1246 at the time of diagnosis than patients with positive N-stage. Studies describe an oncogenic effect of miR-1246 in the majority of tumor types, including LSCC and OSCC, but miR-1246 also appears to play a tumor-suppressive role in lung cancer, hepatocellular carcinoma, and cervical cancer [33]. EV-miR-1246, its molecular effects in OPSCC, and its correlation with lymph metastasis should be continuously explored, as possible pleiotropic properties of this EV-miRNA should not be overlooked.

Our study on the EV expression of miRNAs depending on the form of therapy and including a follow-up examination is among the first. In the HNSCC cohort, the key factor resulting in different expression of EV-miR-181a and -26a depending on therapy modality appears to be the use of chemotherapy. In vitro studies with HNSCC cell lines showed deregulation of various miRNAs upon exposure to cisplatin; among others, miR-181a was downregulated [34]. The effect of cisplatin on miR-181a and the effect of miR-181a in the HNSCC cell lines reflect the pleiotropic properties of the microRNA. Yet the balance between oncogenic and tumor suppressive effects could yet already differ between the localizations of the tumors summarized under HNSCC. Nevertheless, these results indicate potential for EV-miR-181a as a marker for therapy monitoring, independent of its actual molecular effect. Further research on EV-miR-181a as a therapeutic marker in relation to the different tumor entities of HNSCC and the type of therapy could further elucidate the potential of EV-miR-181a in monitoring. Given that various immune cells package tumor-suppressive microRNAs in EVs [9], the effect of chemotherapy on miRNAs in EVs in HNSCC patients could be of particular interest in the search of a monitoring marker. Immune cell-derived exosomes (IEXs) are already being researched as biomarkers for the immunocompetence of the patient with regard to the tumor disease and are an interesting and complementary approach to research on TEX [11].

Regarding a change depending on the therapy modality of miR-26a in tissue, free in serum, or in EVs, there is currently a lack of comparative literature.

As in the overall cohort, in the OPSCC sub-analysis, EV-miR-181a and EV-miR-26a were differently expressed depending on the therapy applied. However, in OPSCC, the performance of surgery and adjuvant therapy seems to make the difference. The changes in EV-miR-181a and EV-miR-26a seen in the OPSCC collective are among the first results documented, and an interpretation is therefore difficult; nonetheless, the results support further research into the suitability of EV-mir181a and -26a for therapy monitoring.

Our study is limited by the observed follow-up period of 12 months on average, being aware that possible recurrence after this period cannot be considered. Furthermore, the EV-miRNA expression was only examined at two points throughout the process of diagnosis, treatment, and follow-up, and especially potentially dysregulated expression right after therapy was not determined.

A challenge in the search for biomarkers for HNSCC is the heterogeneity of the tumors, as, despite the common epithelial origin and close anatomical proximity, some investigated miRNAs either act tumor-suppressive or oncogenic depending on the localization. For this reason, further research on HNSCC and EV-miRNA is necessary, which separately considers the localization.

## 5. Conclusions

In summary, the changes in EV-miR-21 found in this study support the existing literature on its potential as a tumor marker and, at the same time, supplement the still basic knowledge about EV-miR-181a and EV-miR-let-7a in HPV dependence. With an increasing proportion of HPV-associated HNSCC, these two miRNAs should also be further investigated regarding their suitability for the early detection of an HPV association. Additionally, new findings on EV-miR-181a and -26a correlating with the therapy modality either in the HNSCC and in the OPSCC cohort each give reason to investigate both EV-microRNAs as biomarkers for therapy monitoring.

## Figures and Tables

**Figure 1 cancers-16-03793-f001:**
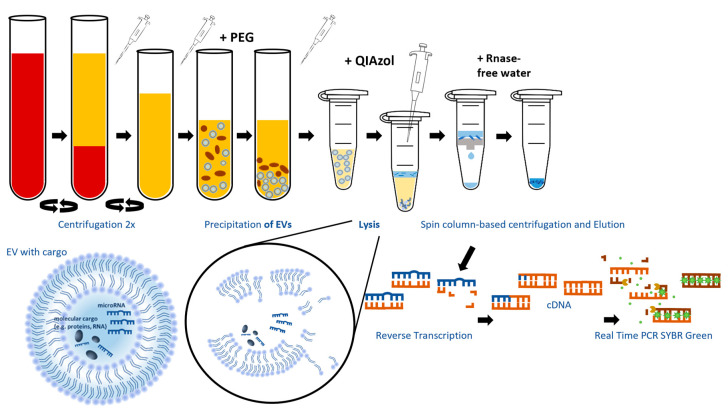
Summary of the isolation and expression analysis of the EV-miRNA.

**Figure 2 cancers-16-03793-f002:**
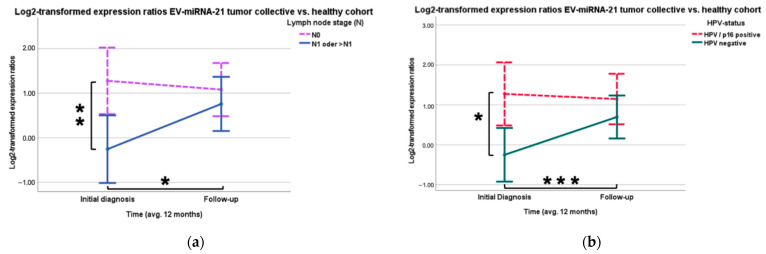
Higher EV-miR-21 expression in patients with negative lymph node status and in patients with HPV/p16 positive tumors. (**a**) EV-miR-21 expression of the tumor cohort in relation to the lymph node involvement. Significantly higher expression in N0 patients (*p* = 0.005) and significantly increased expression in N1 and >N1 patients over the course of 12 months (*p* = 0.032) (* *p* < 0.05, ** *p* < 0.01). (**b**) EV-miR-21 expression of the tumor cohort in relation to HPV status. Significantly higher expression in HPV/p16 positive tumor patients (*p* = 0.003) and significantly increased expression in patients with HPV/p16 negative tumors over the course of 12 months (*p* < 0.001) (* *p* < 0.05, *** *p* < 0.001).

**Figure 3 cancers-16-03793-f003:**
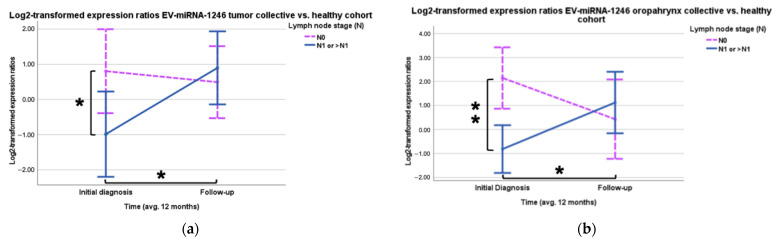
(**a**) EV-miR-1246 expression ratio of the tumor cohort in relation to the lymph node stage. Significantly higher expression in patients with N0 (*p* = 0.035) and significant increase of miR-1246 expression in patients with lymph node metastasis (*p* = 0.029) (* *p* < 0.05). (**b**) EV-miR-1246 expression ratio of the OPSCC cohort in relation to the lymph node stage. Significantly higher expression in patients with N0 (*p* = 0.002) and significant increase over 12 months of EV-miR-1246 expression in patients with lymph node metastasis (*p* = 0.049) (* *p* < 0.05, ** *p* < 0.01).

**Figure 4 cancers-16-03793-f004:**
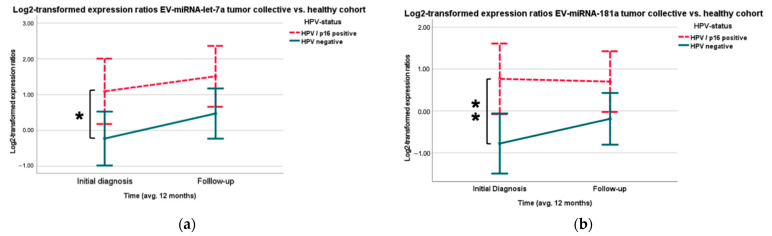
Higher expression of miR-let-7a and—181a in HPV/p16 positive patients. (**a**) Significant differences between the expression ratios of EV-miR-let-7a in patients with and without HPV association (*p* = 0.028) (* *p* < 0.05). (**b**) Significantly higher EV-miR-181a expression in patients with HPV/p16 positive tumors in the HNSCC cohort (*p* = 0.005) (** *p* < 0.01).

**Figure 5 cancers-16-03793-f005:**
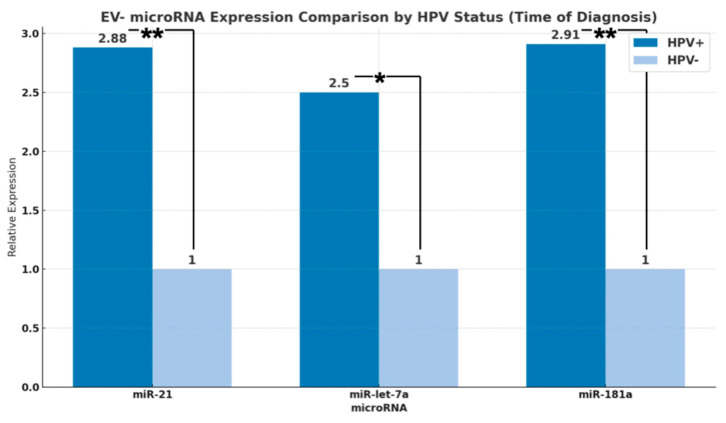
Relative EV expression ratios of miR-21, -let-7a, and -181a comparing the HPV association of the tumor cohort. Significantly higher expression could be seen in HPV/p16 positive tumor patients at initial diagnosis in contrast to HPV/p16 negative patients (* *p* < 0.05, ** *p* < 0.01).

**Figure 6 cancers-16-03793-f006:**
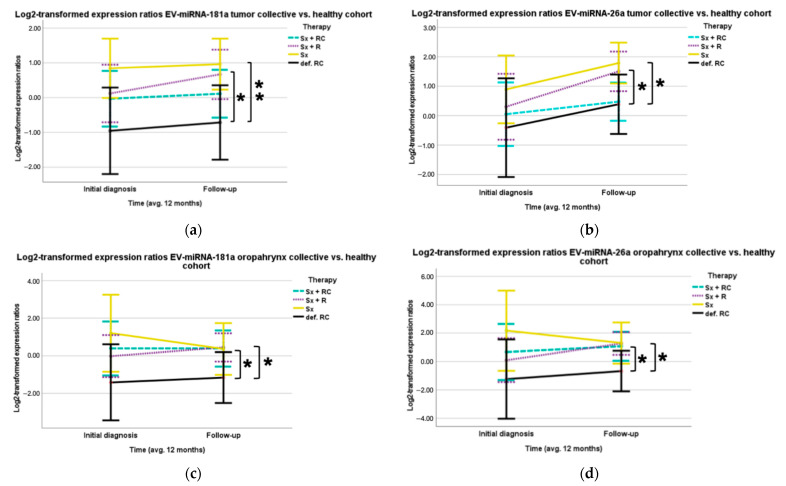
Different expression levels of EV-miR-181a and -26a in relation to therapy modality applied. (**a**,**b**) EV-miR-181a and EV-miR-26a expression ratios of the HNSCC cohort comparing the therapy applied (* *p* < 0.05, ** *p* < 0.01). (**a**) Significantly higher expression of EV-miR-181a in the surgery (*p* = 0.008) and surgery and radiotherapy (*p* = 0.02) groups compared to definitive radio-chemotherapy at time of follow-up. (**b**) The surgery (*p* = 0.017) and the surgery and radiotherapy (*p* = 0.044) groups showed significantly higher EV-miR-26a levels at follow-up. (**c**,**d**) EV-miR-181a and EV-miR-26a expression ratios of the OPSCC cohort comparing therapy applied (* *p* < 0.05, ** *p* < 0.01). (**c**) At follow-up, EV-miR-181a is significantly higher in the groups with surgery and radio-chemotherapy (*p* = 0.046) and surgery and radiation (*p* = 0.026) than in definitive radio-chemotherapy. (**d**) EV-miR-26a is significantly higher in the groups with surgery and radio-chemotherapy (*p* = 0.035) and in surgery and radiation (*p* = 0.013) than in definitive radio-chemotherapy at follow-up.

**Table 1 cancers-16-03793-t001:** TNM stages and localization of the tumor in the HNSCC patients. Absolute numbers and no. (%) of *n* = 49 *. NPSCC = nasopharyngeal squamous cell carcinoma, OPSCC = oropharyngeal squamous cell carcinoma, OSCC = oral squamous cell carcinoma, HPSCC = hypopharyngeal squamous cell carcinoma, LSCC = laryngeal squamous cell carcinoma. * one patient with CUP syndrome, therefore without complete staging.

Characteristics	NPSCC	OPSCC	OSCC	HPSCC	LSCC	Total
T-Stage						
T1	0	5 (10)	0	1 (2)	5 (10)	11 (22)
T2	0	6 (12)	2 (4)	1 (2)	7 (14)	16 (33)
T3	1 (2)	8 (16)	2 (4)	1 (2)	3 (6)	15 (31)
T4	0	3 (6)	0	2 (4)	2 (4)	7 (14)
N-Stage						
N0	0	11 (22)	2 (4)	2 (4)	11 (22)	26 (53)
N1	0	6 (12)	0	1 (2)	1 (2)	8 (16)
N2	1 (2)	5 (10)	0	2 (4)	5 (10)	13 (27)
N3	0	0	2 (4)	0	0	2 (4)
M-Stage						
M0	0	22 (45)	4 (8)	5 (10)	17 (35)	48 (98)
M1	1 (2)	0	0	0	0	1 (2)
UICC						
I	0	7 (14)	0	1 (2)	4 (8)	12 (24)
II	0	5 (10)	2 (4)	1 (2)	4 (8)	12 (24)
III	1 (2)	6 (12)	0	0	3 (6)	10 (20)
IV	0	4 (8)	2 (4)	3 (6)	6 (12)	15 (31)
Total	1 (2)	22 (45)	4 (8)	5 (10)	17 (35)	49 *

**Table 2 cancers-16-03793-t002:** Clinical parameters and localization of the tumor in the HNSCC patients. Absolute numbers and no. (%) of *n* = 50.

Characteristics	NPSCC	OPSCC	OSCC	HPSCC	LSCC	Total
Viral status						
p16+, HPV+	0	11 (22)	0	2 (4)	1 (2)	14 (28)
p16-, HPV-	0	11 (22)	4 (8)	3 (6)	16 (32)	34 (68)
EBV+	1 (2)	1 (2)	0	0	0	2 (4)
Therapy ^1^						
Sx + RC	0	8 (16)	0	0	6 (12)	14 (28)
Sx + R	0	9 (18)	0	1 (2)	0	10 (20)
Sx	0	4 (8)	3 (6)	3 (6)	11 (22)	21 (42)
RC	1 (2)	2 (4)	1 (2)	1 (2)	0	5 (10)
Tobacco						
Smoker	0	9 (18)	4 (8)	3 (6)	10 (20)	26 (52)
Former smoker	0	8 (16)	0	2 (4)	6 (12)	16 (32)
No	1 (2)	6 (12)	0	0	1 (2)	8 (16)
Alcohol						
Active	0	14 (28)	3 (6)	3 (6)	9 (18)	29 (58)
Former C2-abuse	0	3 (6)	0	0	1 (2)	4 (8)
No	1 (2)	6 (12)	1 (2)	2 (4)	7 (14)	17 (34)
Recurrence						
Recurrence-free	0	21 (42)	2 (4)	2 (4)	11 (22)	36 (72)
Recurrence ^2^	1 (2)	2 (4)	2 (4)	3 (6)	6 (12)	14 (28)
Total	1 (2)	23 (46)	4 (8)	5 (10)	17 (34)	50

^1^ Sx = surgery, RC = combined radiation and chemotherapy, R = adjuvant radiation. ^2^ includes recurrence, stable disease, and progressive disease at 12 months follow-up examination.

## Data Availability

The original contributions presented in this study are included in the article/Appendix A; further inquiries can be directed to the corresponding author/s.

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
