# Peer review of "Extracellular Vesicle microRNAs as Possible Liquid Biopsy Markers in HNSCC—A Longitudinal, Monocentric Study"

_cancers, 2024, doi:10.3390/cancers16223793_

Round 1

Reviewer 1 Report

Comments and Suggestions for Authors

The authors report that EV-miRNAs may be promising biomarkers for the diagnosis and treatment monitoring of HNSCC. While reviewing the manuscript, I came across some questions and corrections.

1) In this study, serum EV recovery is extremely important. Evidence of collected EV, such as particle size, particle distribution, and transmembrane protein expression pattern, must be submitted. Furthermore, what was the shape of the EVs released by each cancer cell? It has been suggested that the shape of EVs differs depending on the type of cell releasing them, so this information may also be used as a cancer-specific biomarker.

2) Additionally, information on the purity and quality of the recovered RNA is also missing.

3) Why is regression analysis being used for the statistical analysis of EV miRNA expression at the time of initial diagnosis? Wouldn't a t-test be sufficient? Wouldn't a scatter plot be better for presenting regression coefficients?

4) Since the difference in miRNA expression due to the progression of cancer disappears as time passes from the initial diagnosis, does this mean that they will no longer be useful as biomarkers depending on the timing of diagnosis?

Author Response

Dear Sir or Madame,

Thank you very much for the review and the important questions and annotations that you have regarding our study. We wish to answer all of them in the following point-by-point:

Review: The authors report that EV-miRNAs may be promising biomarkers for the diagnosis and treatment monitoring of HNSCC. While reviewing the manuscript, I came across some questions and corrections.

1) In this study, serum EV recovery is extremely important. Evidence of collected EV, such as particle size, particle distribution, and transmembrane protein expression pattern, must be submitted. Furthermore, what was the shape of the EVs released by each cancer cell? It has been suggested that the shape of EVs differs depending on the type of cell releasing them, so this information may also be used as a cancer-specific biomarker.

Response: Thank you for bringing up this important point. When deciding upon the study design and methods in 2020, we wanted to use a method that is easily applicable in clinical routines and very sensitive to detect EVs. We chose the miRCURY Exosome Serum/Plasma Kit (Qiagen, Hilden, Germany) because it met our requirements at this point. Helwa et al. used Nanoparticle Tracking Analysis and TEM imaging to ascertain quantity and size of EVs recovered by miRCURY Exosome Serum/Plasma Kit (A Comparative Study of Serum Exosome Isolation Using Differential Ultracentrifugation and Three Commercial Reagents, Helwa et al., 2017) that showed consistently particles within the expected size of 40-150 nm and carrying CD63 and CD9 recovered by the miRCURY Exosome Serum/Plasma Kit. Qiagen guarantees to provide an EV-isolation kit, that has a high yield and purity of the isolates. Most crucial to our decision was the comparatively ease of implementation of our chosen method in the clinical routines, as we wanted to apply a sensitive and feasible EV-recovery, too.

Therefore, we added the following sentence on page 4, line 155 - 157, paragraph 2.3.: “Demonstration of suitability of the kit regarding size, surface markers and quantity of recovered EVs is given by Helwa et al. [22]. Particle size recovered by the kit ranges in between 40-150nm.” The reference was added to the bibliography.

2) Additionally, information on the purity and quality of the recovered RNA is also missing.

Response: Thank you for this feedback. We agree with this comment. Therefore, we added the following sentence on page 4, line 165/166, paragraph 2.4.: “Isolated RNA was of high purity (OD 260/ OD 280 = 1.8 – 2.0) and moderate to high quality (RIN ≥ 6).”

3) Why is regression analysis being used for the statistical analysis of EV miRNA expression at the time of initial diagnosis? Wouldn't a t-test be sufficient? Wouldn't a scatter plot be better for presenting regression coefficients?

Response: Thank you for pointing this out. We decided to use general linear models to create a prognostic model that includes all clinical features investigated for each EV-miRNA. We found this method as appropriate as the patients received an intervention (UICC depending oncological therapy). Using this method, we were able to analyze the effect of the clinical parameters at the time of diagnosis and at the time of the follow-up separately. In the following, we used t-tests to investigate the changes of the EV-miRNA-expression in relation to the different clinical features to describe the dynamic of the expression over the investigated time (e.g. Lymph node involvement at time of diagnosis and at the follow-up examination). Your question is of importance as we did not mention the use of the t-tests but presented the results in the figures by marking changes of significances as horizontal brackets. We therefore improve the mentioning of the tests on page 5, line 216-218, paragraph 2.7.: “Using paired t-tests, the expression ratios of the six miRNAs in relation to the clinical parameters examined at the time of surgery and at the follow-up (avg. 12 months) were analyzed for significant changes over this period.”  

4) Since the difference in miRNA expression due to the progression of cancer disappears as time passes from the initial diagnosis, does this mean that they will no longer be useful as biomarkers depending on the timing of diagnosis?

Response: Thank you for this annotation. We understand that this question aims to investigate the suitability of the EV-miRNA in early diagnosis versus late diagnosis of HNSCC. We have in fact also investigated the expression of the EV-miRNA in relation to the UICC stage. The results are, however, not part of our article. We want to emphasize the fact that in between the time of diagnosis and the follow-up the patients received therapy. We therefore do not have any findings regarding the changes of expression of EV-miRNA in patients with progressive disease, or not intervened cancer progress. Our interpretation of the disappearing differences is the steady normalization of the expression of the EV-miRNAs (always in comparison to the healthy cohort) as the cancer is (successfully for now) treated.

Sincerely yours,

Maximilian Oberste

Reviewer 2 Report

Comments and Suggestions for Authors

In the current manuscript, the clinical utility of the extracellular vesicles microRNA-21, -1246, -let-7a, -181a and -26a were evaluated in a total of 50 head and neck squamous cell carcinoma (HNSCC) patients before and one year after surgery/treatment. Mir-21, and -1246 showed higher expression in patients with no lymph node metastasis compared to those with positive lymph nodes. Moreover, HPV/p16+ patients showed higher mir-21, -7, and 181a expression compared to HPV Neg HNSCC patients. Whereas In OPSCC patients, it was only mir-1246 which showed high expression in lymph node Neg patients. In terms of monitoring surgery, mir-181a showed lowest expression in patients who went for definitive Radio-chemotherapy (def-RC) compared to all other groups. There was no striking change in the expression of mir-181 in the follow up test. Similarly, mir-26a showed similar pattern of expression at initial diagnosis and only surgery and surgery + radio groups showed around one-fold increase at follow up.

Finding liquid biopsy-based biomarkers able to monitor treatment efficiency is a hot topic and an active area of research. However, the current study did not provide sufficient analyses for potential biomarker evaluation. Please find my detailed comments below:

Major concerns:

1-     For biomarker assessment, survival analyses are the golden standard. Since the follow up period is too short, one cannot draw a conclusion about any of the markers under study.

2-     Numbers of patients groups used throughout the study  are too small. It gets even smaller for subgroup analyses such as testing the OPSCC and testing correlation with different treatment modalities. Some groups are represented with only 2 patients.

3-     Criteria for deciding the treatment plan should be included. One would predict that lymph node involvement is a major factor on treatment decision. Since multiple miRs showed strong association with lymph node status, it is important to test whether the associations between miRs and treatment modalities are independent of lymph node status.

Minor Concerns:
1- Page 8 (line 282:286), the sentence is not clear and needs rephrasing.

2- Page 6 (Line 236), “with HPV/p16 tumors”, this should be “HPV/P16 Neg tumors”??

3-Page 8 (Line 357:358): “the results observed by us are in line with 357 several studies that could not show a significant correlation between the expression of 358 miR-21 in tumor tissue and the N-stage in OPSCC [29, 30]”. However, authors did not present data for the association between MiR-21 and lymph node status in OPSCC. Rather, they showed in HNSCC that higher expression of miR-21 is associated with Lymph Node Neg status which was not discussed.

Comments on the Quality of English Language

English language is fine but needs few edits.

Author Response

Dear Sir or Madame,

We want to thank you for your report and the important questions put forward in your review. We hope to answer appropriately to your questions in the following: 

Review: In the current manuscript, the clinical utility of the extracellular vesicles microRNA-21, -1246, -let-7a, -181a and -26a were evaluated in a total of 50 head and neck squamous cell carcinoma (HNSCC) patients before and one year after surgery/treatment. Mir-21, and -1246 showed higher expression in patients with no lymph node metastasis compared to those with positive lymph nodes. Moreover, HPV/p16+ patients showed higher mir-21, -7, and 181a expression compared to HPV Neg HNSCC patients. Whereas In OPSCC patients, it was only mir-1246 which showed high expression in lymph node Neg patients. In terms of monitoring surgery, mir-181a showed lowest expression in patients who went for definitive Radio-chemotherapy (def-RC) compared to all other groups. There was no striking change in the expression of mir-181 in the follow up test. Similarly, mir-26a showed similar pattern of expression at initial diagnosis and only surgery and surgery + radio groups showed around one-fold increase at follow up.

Finding liquid biopsy-based biomarkers able to monitor treatment efficiency is a hot topic and an active area of research. However, the current study did not provide sufficient analyses for potential biomarker evaluation. Please find my detailed comments below:

Major concerns:

1-     For biomarker assessment, survival analyses are the golden standard. Since the follow up period is too short, one cannot draw a conclusion about any of the markers under study.

Response: Thank you for bringing up this concern. When deciding upon the study design, we chose a prospective study design as we wanted to find further knowledge on the EV-miRNA-expression of the HNSCC patients at time of diagnosis and after they were treated. Lin et al. investigated the different expression levels of several miRNAs (among them miR181a) before and after therapeutical intervention in patients with esophageal squamous cell carcinoma (Lin Z, Chen Y, Lin Y, Lin H, Li H, Su X, et al. Potential miRNA biomarkers for the diagnosis and prognosis of esophageal cancer detected by a novel absolute quantitative RT-qPCR method. Scientific reports 2020;10:20065). There are only few longitudinal studies that examine EV-miRNA-expression in HNSCC, and we would like to add to the existing discussion about the suitability of the examined EV-miRNAs, as they seem to partially act differently in HNSCC compared to others and already better studied neoplasia. With our sub-analysis we like to focus on OPSCC specifically as there is heterogenous literature on how the EV-miRNAs investigated, act in OPSCC to HNSCC in contrast.

2-     Numbers of patients groups used throughout the study are too small. It gets even smaller for subgroup analyses such as testing the OPSCC and testing correlation with different treatment modalities. Some groups are represented with only 2 patients.

Response: The question presented regarding the size of our cohort is reasonable. We decided on the cohort size relying on similar published studies with EVs or “exosomes” in HNSCC patients that consists of a comparable number of patients. The group of patients with OPSCC that received definitive chemoradiation consists of 2 patients. However, we decided to include the results of the sub-analysis with the OPSCC patients regarding the therapy modality, as it reinforces the results of the overall tumor cohort. If there had not been any results in the investigation of the therapy modality applied and the EV-miRNAs 181a and 26a in the general tumor cohort, we would not have put these results forward. However, in this case we think that the results of the sub-analysis only reinforce the results of the overall group and emphasize the potential of EV-miR-181a and 26a in therapy monitoring. We agree that in the next step the study design should include a higher number of patients to reinvestigate the presented results, maybe including a longer follow-up period and find significant correlation between the EV-miRNAs investigated and recurrence of disease.

3-     Criteria for deciding the treatment plan should be included. One would predict that lymph node involvement is a major factor on treatment decision. Since multiple miRs showed strong association with lymph node status, it is important to test whether the associations between miRs and treatment modalities are independent of lymph node status.

Response: Thank you for this critical and important point. The decision upon the treatment plan was made by the interdisciplinary tumor board (page 3, line 134, paragraph 2.1.) and followed existing guidelines in Germany (S3-guideline for laryngeal cancer, oral cancer and oropharyngeal/hypopharyngeal cancer). The treatment modality is chosen depending on the UICC stage (including T, N and M-stages). However, the HPV-status impacts the UICC stage in OPSCC, too. We investigated the impact of the clinical features (TNM, tobacco, alcohol, UICC, therapy modality) each by using the general linear models that allow to calculate the individual impact of each feature. We found several clinical features that had significant impact on the expression levels of miR-181a (e.g. tumor size, UICC), therefore the lymph node involvement did not stand alone. A higher UICC stage correlated with higher EV-expression of miR-181a, one would therefore expect the group that received definitive radio-chemotherapy (often used in patients with unresectable tumor of bigger size) to express miR-181a higher as well, maybe even at the time of the follow-up. However, the expression levels were both higher in the groups with patients that were treated with surgery and that were treated with surgery and adjuvant radiotherapy (Fig. 6a).

Minor Concerns:
1- Page 8 (line 282:286), the sentence is not clear and needs rephrasing.

Response: We like to rephrase the sentence: “Compared to the patients who had been treated with definitive combined radio-chemotherapy (n = 4), the group treated with surgery and adjuvant radiotherapy (n = 10) showed 2.61 times higher (Log2 regression coefficient: 1.383, p = 0.02) and 3.2 times higher (Log2 regression coefficient: 1.679, p = 0.008) EV-expression levels of miR-181a in the group treated only by surgery (n = 20) (Fig. 6 (a)) (Figure A5).” to the following, now on page 9, lines 295-302, paragraph 3.2.: “In comparison to the patients treated with definitive combined radio-chemotherapy (n = 4), the group that received surgery and adjuvant radiotherapy (n = 10) showed 2.61 times higher (Log2 regression coefficient: 1.383, p = 0.02) EV-expression levels of miR-181a at the time of follow-up (Fig. 6 (a)). Similarly, the patients that received only surgery (n = 20) showed 3.2 times higher (Log2 regression coefficient: 1.679, p = 0.008) EV-expression levels of miR-181a in comparison to the group of patients that were treated with surgery and adjuvant radiotherapy (n = 10) at the time of follow-up (Fig. 6 (a)).“

2- Page 6 (Line 236), “with HPV/p16 tumors”, this should be “HPV/P16 Neg tumors”??

Response: Thank you very much for this comment, you are right, we used to abbreviate HPV/p16 negative tumors with “HPV/p16-“ and apparently did not include either ‘–‘ or the correct term ‘negative’ in our manuscript. Thank you for bringing this to our attention, we will certainly change this. To prevent misunderstandings elsewhere, the abbreviation has been improved and changed in all passages in the manuscript.

3-Page 8 (Line 357:358): “the results observed by us are in line with 357 several studies that could not show a significant correlation between the expression of 358 miR-21 in tumor tissue and the N-stage in OPSCC [29, 30]”. However, authors did not present data for the association between MiR-21 and lymph node status in OPSCC. Rather, they showed in HNSCC that higher expression of miR-21 is associated with Lymph Node Neg status which was not discussed.

Response: Thank you for bringing this to our attention, as we need to make sure to present a coherent argumentation. We need to add the information that no results of significance were found when investigating the correlation between the lymph node stage and EV-mir-21 in OPSCC. In HNSCC in general, e.g. in OSCC, as mentioned in the discussion (page 11, line 375 -376, paragraph 4, cited study by Li et al.), miR-21 expression often correlates with lymph node involvement. As we want to work out this difference in OPSCC, we need to clarify this in our discussion, thank you for making this point:

Page 11, lines 375-376, paragraph 4: “as we neither could find significant correlation of EV-miR-21 and N-stage in OPSCC”

Sincerely yours,

Maximilian Oberste

Reviewer 3 Report

Comments and Suggestions for Authors

The authors here reported the potential of applying EV microRNAs as the liquid biopsy markers in HNSCC. Through the detection, some miRNAs were validated to be correlated with HNSCC prognosis. This work provides some clues and instructions of exploiting some EV-derived miRNAs as the biomarkers of HNSCC. I have some points to discuss with the authors to further improve the quality of the current manuscript for publication in Cancers.

1. An illustrative Figure showing the whole route of detecting EV-miRNAs markers from patients using liquid biopsy is needed for easier readability.

2. The Figure 1 is not fully showing the specific principle of miRNA detection. Maybe the principle of miRNA detection is needed to be shown. Stem-loop RT PCR? The expression level of all these detected miRNAs and Spike-in control genes should be shown in a supplementary table.

3. What is the diameter of EVs extracted from these patients? Is these biomarker miRNAs all detected in the test patients?

4.  In line 170-171, is the 0,5, 1,8, 0,5wrongly written? The similar thing like “3,0” in line 185 is also needed to be noted. Please confirm them.

5. The selection of miR-16 as an endogenous control can be somewhat described in the text.

6.   In the caption of Figures 2, 3, 4, 6, a simple conclusion title is needed.

Author Response

Dear Sir or Madame,

Thank you very much for taking the time to review this manuscript. Please find our detailed responses below and the corresponding revisions highlighted in the re-submitted files.

Review: The authors here reported the potential of applying EV microRNAs as the liquid biopsy markers in HNSCC. Through the detection, some miRNAs were validated to be correlated with HNSCC prognosis. This work provides some clues and instructions of exploiting some EV-derived miRNAs as the biomarkers of HNSCC. I have some points to discuss with the authors to further improve the quality of the current manuscript for publication in Cancers.

  1. An illustrative Figure showing the whole route of detecting EV-miRNAs markers from patients using liquid biopsy is needed for easier readability.

Response: Thank you for your constructive feedback, we provide an improved figure showing the whole route of detecting EV-miRNA markers from our patients.

  1. The Figure 1 is not fully showing the specific principle of miRNA detection. Maybe the principle of miRNA detection is needed to be shown. Stem-loop RT PCR? The expression level of all these detected miRNAs and Spike-in control genes should be shown in a supplementary table.

Response: Thank you again for making this point, the readability of our article is of great importance to us and we  provide an improved figure 1 now to show the specific principle of miRNA detection completely. We used miCURY LNA miRNA SYBR Green PCR System (Qiagen) for miRNA detection.  If you wish, we can of course include all the tables showing the miRNAs and Spike-in control genes in the supplements.

  1. What is the diameter of EVs extracted from these patients? Is these biomarker miRNAs all detected in the test patients?

Response: Thank you for pointing this out. We used the miRCURY Exosome Serum/Plasma Kit (Qiagen, Hilden, Germany). . Helwa et al. used Nanoparticle Tracking Analysis and TEM imaging to ascertain quantity and size of EVs recovered by miRCURY Exosome Serum/Plasma Kit (A Comparative Study of Serum Exosome Isolation Using Differential Ultracentrifugation and Three Commercial Reagents, Helwa et al., 2017) that showed consistently particles within the expected size of 40-150 nm and carrying CD63 and CD9 recovered by the miRCURY Exosome Serum/Plasma Kit. Qiagen guarantees - with that literature proof - an EV-isolation kit, that has a high yield and purity of the isolates.

We added the following sentence on page 4, line 155 - 157, paragraph 2.3.: “Demonstration of suitability of the kit regarding size, surface markers and quantity of recovered EVs is given by Helwa et al. [22]. Particle size recovered by the kit ranges in between  40-150nm.” The reference was added to the bibliography.

Regarding your second question we understand that ‘test patients’ are patients with HNSCC. In this case, yes, all EV-miRNAs were detected in the serum of the 50 investigated patients.

  1.  In line 170-171, is the “0,5”, “1,8”, “0,5” wrongly written? The similar thing like “3,0” in line 185 is also needed to be noted. Please confirm them.

Response: Thank you very much for your attention to detail, you are right, and we need to change this to the standard way of writing decimal numbers. We will apply this to the article immediately.

  1. The selection of miR-16 as an endogenous control can be somewhat described in the text.

Response: You are correct to propose to explain our reason to choose miR-16 as our housekeeper and we will elaborate on page 5, lines 196-197, paragraph 2.6.: MiRNA-16 is often used as endogenous reference in similar studies on miRNA in HNSCC, e.g. by Piao et al. in OSCC [23].” The reference was added to the bibliography.

  1.   In the caption of Figures 2, 3, 4, 6, a simple conclusion title is needed.

Response: Thank you, we complete our figure captions with simple conclusion titles.

Sincerely yours,

Maximilian Oberste

Round 2

Reviewer 1 Report

Comments and Suggestions for Authors

The authors have responded sincerely to the reviewers' comments.

Author Response

Review: The authors have responded sincerely to the reviewers' comments.

Response: Thank you very much for your quick processing of the review!

Reviewer 2 Report

Comments and Suggestions for Authors

I thank the authors for clearly addressing my concerns. I have no further comments.

Author Response

Review: I thank the authors for clearly addressing my concerns. I have no further comments.

Response: Thank you very much for your quick processing of the review!

Reviewer 3 Report

Comments and Suggestions for Authors

All my concerns have been well resolved. For replicable data reference by other teams,all the data showing the Ct values from RT-qPCR are still necessary to be provided as the supplementary table. 

Author Response

Report 3

Review: All my concerns have been well resolved. For replicable data reference by other teams, all the data showing the Ct values from RT-qPCR are still necessary to be provided as the supplementary table.

Response:

Dear Sir or Madame,

Thank you for pointing this out. We agree with your comment. Therefore, we have provided a supplementary table listing all Ct-values of investigated microRNAs miR-21, miR-1246, miR-200c, miR-let-7a, miR-181a, miR-26a and internal controls (Uni-Spike-In). The miRNAs were analyzed three times, and the average value was calculated (avg. miR). At the time of the follow-up examination the method was applied identically, and the results are each marked with “FU” (Follow-up).

We added the reference to our supplementary table in the main text on page 4, line 164-166, paragraph 2.4: “A table listing the Ct-values of the investigated miRNAs and Uni-Spike-In controls can be found in the supplements (Table S1)and on page 12, line 433-435, Supplementary materials: Table S1: Supplementary Table listing all Ct-values of investigated microRNAs miR-21, miR-1246, miR-200c, miR-let-7a, miR-181a, miR-26a and internal controls (Uni-Spike-In).

Sincerely yours,

Maximilian Oberste